



# The Roland von Glasow Air-Sea-Ice Chamber (RvG-ASIC): an experimental facility for studying ocean/sea-ice/atmosphere interactions

Max Thomas[1], James France[1,2,3], Odile Crabeck[1], Benjamin Hall[4], Verena Hof[5], Dirk Notz[5,6], Tokoloho Rampai[4], Leif Riemenschneider[5], Oliver Tooth[1], Mathilde Tranter[1], and Jan Kaiser[1]

[1]Centre for Ocean and Atmospheric Sciences, School of Environmental Sciences, University of East Anglia, UK, NR4 7TJ
[2]British Antarctic Survey, Natural Environment Research Council, Cambridge CB3 0ET, UK
[3]Department of Earth Sciences, Royal Holloway, University of London, Egham TW20 0EX, UK
[4]Chemical Engineering Deptartment, University of Cape Town, South Africa
[5]Max Planck Institute for Meteorology, Hamburg, Germany
[6]Center for Earth System Research and Sustainability (CEN), University of Hamburg, Germany

**Correspondence:** Jan Kaiser (j.kaiser@uea.ac.uk)

**Abstract.** Sea ice is difficult, expensive, and potentially dangerous to observe in nature. The remoteness of the Arctic and Southern Oceans complicates sampling logistics, while the heterogeneous nature of sea ice and rapidly changing environmental conditions present challenges for conducting process studies. Here, we describe the Roland von Glasow Air-Sea-Ice Chamber (RvG-ASIC), a laboratory facility designed to reproduce polar processes and overcome some of these challenges. The RvG-

ASIC is an open-topped 3.5 m$^3$ glass tank housed in a coldroom (temperature range: -55 to +30 $^{\text{o}}$C). The RvG-ASIC is equipped with a wide suite of instruments for ocean, sea ice, and atmospheric measurements, as well as visible and UV lighting. The infrastructure, available instruments, and typical experimental protocols are described.

To characterise some of the technical capabilities of our facility, we have quantified the timescale over which our chamber exchanges gas with the outside, $\tau_l = (0.66 \pm 0.07)$ days, and the mixing rate of our experimental ocean, $\tau_m = (4.2 \pm 0.1)$

minutes. Characterising our light field, we show that the light intensity across the tank varies by less than 10 % near the centre of the tank but drops to as low as 60 % of the maximum intensity in one corner. The temperature sensitivity of our light sources over the 400 nm to 700 nm range (PAR) is $(0.028 \pm 0.003)$ W m$^{-2}$ $^{\text{o}}$C$^{-1}$, with a maximum irradiance of 26.4 W m$^{-2}$ at 0 $^{\text{o}}$C; over the 320 nm to 380 nm range, it is $(0.16 \pm 0.1)$ W m$^{-2}$ $^{\text{o}}$C$^{-1}$, with a maximum irradiance of 5.6 W m$^{-2}$ at 0 $^{\text{o}}$C.

We also present results characterising our experimental sea ice. The extinction coefficient for PAR varies from 3.7 m$^{-1}$ to 6.1

m$^{-1}$ when calculated from irradiance measurements exterior to the sea ice and from 4.4 m$^{-1}$ to 6.2 m$^{-1}$ when calculated from irradiance measurements within the sea ice. The bulk salinity of our experimental sea ice is measured using three techniques, modelled using a halo-dynamic one-dimensional (1D) gravity drainage model, and calculated from a salt and mass budget. The growth rate of our sea ice is between 2 cm d$^{-1}$ and 4 cm d$^{-1}$ for air temperatures of $(-9.2 \pm 0.9)$ $^{\text{o}}$C and $(-26.6 \pm 0.9)$ $^{\text{o}}$C. The PAR extinction coefficients, vertically integrated bulk salinities, and growth rates all lie within the range of previously reported

comparable values for first-year sea ice. The vertically integrated bulk salinity and growth rates can be reproduced well by a 1D



model. Taken together, the similarities between our laboratory sea ice and observations in nature, and our ability to reproduce our results with a model, give us confidence that sea ice grown in the RvG-ASIC is a good representation of natural sea ice.

*Copyright statement.* TEXT

## 1 Introduction

Sea ice lies at the ocean-atmosphere interface. As such, sea ice mediates the exchange of energy (e.g. Grenfell and Maykut, 1977), momentum (e.g. McPhee et al., 1987), gases (e.g. Gosink et al., 1976), and particles (e.g. May et al., 2016) between the polar oceans and the atmosphere. Sea ice formation provides buoyancy forcing to the underlying ocean (Worster and Rees Jones, 2015). Sea-ice algae inhabit brine inclusions in the sea ice and can reach high concentrations, with nutrients resupplied by gravity drainage (Fritsen et al., 1994).

The remoteness and extremeness typical of the polar oceans makes observing sea ice *in situ* difficult, expensive, and potentially dangerous — both for personnel and equipment. Such logistical challenges are heightened during the initial growth and final melt of sea ice, which are particularly interesting study periods. Also, the heterogeneous nature of sea ice, with important parameters varying over sub-meter scales (Miller et al., 2015) and responding to sub-diurnal fluctuations in air temperature, make conducting process studies in the field difficult. These logistical and scientific difficulties motivate the use of laboratory

grown sea ice to bridge observational gaps.

Existing laboratory facilities vary widely, tending to be designed with specific observations in mind and circumventing specific constraints. The length of sea-ice tanks varies from tens of meters (e.g. Rysgaard et al., 2014; Cottier et al., 1999) to tens of centimetres (e.g. Eide and Martin, 1975; Nomura et al., 2006). Larger tanks minimise edge effects and allow for more samples to be taken. Smaller tanks are cheaper to build and run and have better controlled boundary conditions. The cooling

method used to form artificial sea ice ranges from cold plates (Cox and Weeks, 1975; Eide and Martin, 1975; Wettlaufer et al., 1997) through the controlled atmosphere of a coldroom (Tison et al., 2002; Style and Worster, 2009; Naumann et al., 2012; Marks et al., 2017) to the outdoors (Rysgaard et al., 2014; Hare et al., 2013). Coldplates provide the tightest control over the upper boundary condition but preclude the study of many processes at the upper interface because the experimental system has no atmosphere. There are few chambers in the literature with atmospheres enclosed in a headspace (Nomura et al., 2006),

possibly because, as noted by Loose et al. (2011), the temperature of an enclosed headspace tends to be much warmer than the coldroom in which the experimental system is housed. Most facilities use the entire coldroom as an atmosphere. Tanks have been made of glass (Naumann et al., 2012), plastic (Loose et al., 2009; Eide and Martin, 1975), stainless steel (Shaw et al., 2011), and concrete (Rysgaard et al., 2014). Of these, glass has the advantage of excellent chemical inertness while steel, concrete, and plastic are cheaper and can be more robust. Glass and plastic allow the sea ice to be observed through the

tank sides. Lighting may be placed above a sea-ice tank to provide illumination for radiative transfer experiments (Perovich and Grenfell, 1981; Marks et al., 2017). Finally, sea-ice tanks are either cuboid (e.g. Tison et al., 2002; Naumann et al., 2012;





Rysgaard et al., 2014; Nomura et al., 2006; Wettlaufer et al., 1997), cylindrical (e.g. Cox and Weeks, 1975; Perovich and Grenfell, 1981; Loose et al., 2009; Marks et al., 2017), or quasi two-dimensional Hele-Shaw cells (Eide and Martin, 1975; Middleton et al., 2016). Wave generation is easier in a cuboid tank whereas cylindrical tanks will have simpler edge effects.
Hele-Shaw cells allow visualisation of the internal sea-ice structure.

There are several examples of sea-ice tanks having proved effective in generating and testing hypotheses. As an example, much of our understanding of gravity drainage was produced or refined using sea-ice tank experiments. Laboratory studies traced brine (Cox and Weeks, 1975; Eide and Martin, 1975), visualised brine channels (Niedrauer and Martin, 1979), and provided idealised conditions to evaluate models (Wettlaufer et al., 1997; Notz, 2005) and measurement techniques (Notz et al.,
2005). The understanding, qualitative and quantitative, of processes developed by these studies has led to the development of physically faithful, precise gravity drainage parameterisations (Wells et al., 2011; Griewank and Notz, 2013; Turner et al., 2013; Rees Jones and Worster, 2014). Sea-ice tanks also played a key role in developing models of gas transport in and around sea ice. Laboratory experiments have been used to estimate the magnitude of gas fluxes (Nomura et al., 2006), investigate the processes by which gases are transported (Loose et al., 2009; Kotovitch et al., 2016; Shaw et al., 2011), and to investigate
ocean-atmosphere gas fluxes through breaks in sea-ice cover (Loose et al., 2011; Lovely et al., 2015). For both gas fluxes and gravity drainage, progress was made by integrating laboratory studies with field measurements and modelling (e.g. Zhou et al., 2014; Notz and Worster, 2008). Laboratory sea ice has helped develop our understanding of radiative transfer in sea ice (Perovich and Grenfell, 1981; Marks et al., 2017), frost flower formation (Style and Worster, 2009; Perovich and Richter-Menge, 1994), the thermodynamics of grease ice and nilas (Naumann et al., 2012; De La Rosa et al., 2011), and many other
fields.

Here, we describe and characterise the Roland von Glasow Air-Sea-Ice Chamber (RvG-ASIC) to aid future users when planning experiments. We first describe the facility, the instrumentation available, and typical protocols used to grow artificial sea ice (Section 2). Next, we characterise the air exchange rate of the chamber with the outside, the mixing rate of the tank, and the variability of the light field —— important technical metrics for designing and interpreting experiments (Section 3.1).
We then evaluate how similar our sea ice is to natural sea ice by comparing our experimental sea ice to natural sea ice for three key parameters: the extinction of photosynthetically active radiation, the bulk salinity, and the growth rate (Section 3.2).

## 2    Facility description

### 2.1    Infrastructure

Our artificial ocean is contained in a cuboid, open-topped, glass tank (Figure 1). The internal footprint of the tank is 2.35 m ×
1.35 m and the depth of the tank is 1.17 m. The glass is 25 mm thick, joined at the edges with silicone resin, and reinforced with a metal bracketing bar. A smaller cuboid glass tank (0.5 m × 0.4 m footprint, 1.12 m depth, 12 mm wall thickness) is joined to the main tank, connected by four 100 mm holes (Figure 1). This side tank is never allowed to freeze over entirely and so provides a path for sample lines into the ocean, a path for cables that does not interfere with the sea-ice/atmosphere interface, and a free path for water displaced by volume expansion upon freezing. The main and side tanks have been insulated using 10





cm of Dow 500A floormate foam (0.035 W m$^{-1}$ K$^{-1}$ thermal conductivity) and 10 cm of foil-backed loft insulation. Heating film (12 V, maximum output 220 W m$^{-2}$) was placed outside of the glass and within the insulation. The heating pads and insulation ensure that the dominant cooling of our experimental system is at the exposed upper interface rather than through the tank sides. They also prevent sea ice attaching to our tank walls so that it remains free floating. Two to four pumps (TUNZE stream 6125, maximum flow rate = 12000 l h$^{-1}$) are placed in the main and side tanks to mix the ocean. The pumps are fixed

to the tank walls using magnets and can be positioned as needed.

A lighting rack sits at 1.3 m above the tank base (Figure 1). Eight UV-B lamps (Philips broadband TL 100 W), eight UV-A lamps (Philips Cleo performance 100 W), and eight visible LED strips (Fluence solar max) are spaced evenly over the length of the tank. The visible lights can be individually dimmed from 100 % to 10 % of the full output. Their spectra are shown in Figure 2.

The tank and lights are housed in a coldroom with a temperature range of -55 to +30 ºC. The external footprint of the coldroom is 4.32 m × 4.32 m. The external height is 3.15 m and the internal height is is 2.85 m (Figure 3). The internal surfaces of the coldroom are stainless steel. Six fans recirculate air within the coldroom via a compressor. The coldroom is connected to the outside by double doors, six 10 cm holes for sample/power/data lines, one drainage hole, and one ventilation hole. All of these can be closed, and sample/power/data lines can be fed in through gas tight Roxtec ports. The coldroom is

controlled by a Eurotherm nanodac control panel.

The coldroom is located in an external laboratory. The control boxes for instrumentation, heating, pumps, cameras, and valves are all situated adjacent to the coldroom. Instruments log to an Envidas Ultimate acquisition system software for continuous monitoring of the data. Mains or 18.2 MΩ water supplies (Centra R 200) are available for filling the tank.

## 2.2 Instrumentation

The RvG-ASIC has a suite of instruments for measuring in experimental ocean, sea ice, and atmosphere (Table 1).

### 2.2.1 Ocean

The ocean temperature, $\theta_o$, and conductivity, $\gamma_o$, can be monitored by a temperature and conductivity recorder (Seabird Microcat SBE 37SIP or Valeport mini CT). $\theta_o$ and $\gamma_o$ measurements are converted to a salinity, $S_o$ (g kg$^{-1}$), using either the GSW toolbox (McDougall and Barker, 2011) for representative ocean salt composition or using equations presented in Naumann

et al. (2012) for pure NaCl. We deploy a sonar (Aquascat 1000R) to measure the position of the waterline at the start of the experiment and the position of the base of the sea ice throughout the experiment. We use the GSW toolbox to calculate the speed of sound for a given $S_o$ and $\theta_o$ and scale the raw sonar output to produce an accurate distance measurement. We can deploy a Liquicel 1.7×5.5 MiniModule in our ocean coupled to a LGR GGA-30r-EP OA-ICOS laser spectrometer to measure the concentration of dissolved $CH_4$ and $CO_2$ in the ocean without removing water from the experimental system.



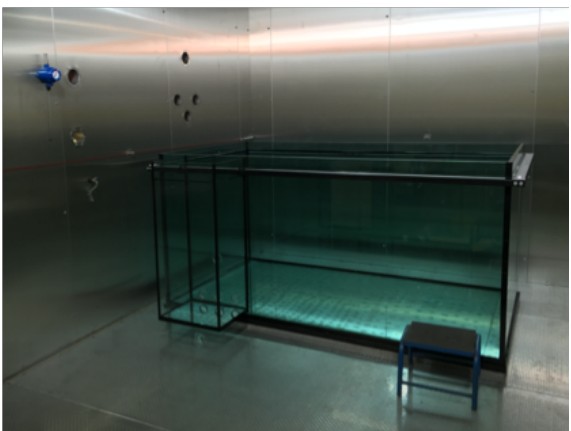

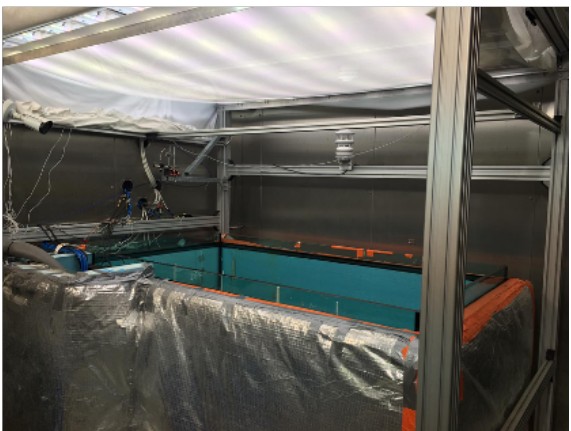

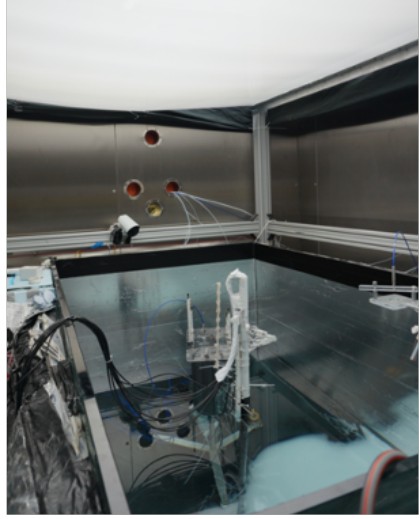

**Figure 1.** The tank just after installation (top), with all the main features in place (middle), and set up for experiments with visible lighting (bottom).





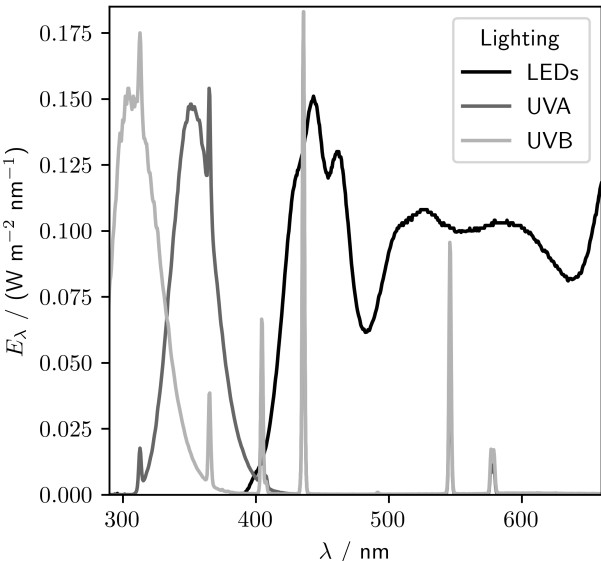

**Figure 2.** Spectral irradiance, $E_\lambda$, for the lights in the RvG-ASIC at 1 m height and room temperature.

**Table 1.** Instruments that are generally available in the RvG-ASIC.

| Instrument | Parameter | Experiment compartment |
|---|---|---|
| Seabird Microcat SBE 37-SIP | conductivity, temperature | ocean |
| DS18B20U digital thermometer | temperature | sea ice / ocean |
| wireharp | liquid fraction | sea ice |
| TCS3472 photodiodes | irradiance, 400 nm to 700 nm | sea ice / ocean / atmosphere |
| ocean optics spectrometer (USB2000+) | spectral irradiance | sea ice / ocean / atmosphere |
| Met-con spectral radiometer | wavelength resolved 280 nm to 650 nm actinic flux | sea ice / ocean / atmosphere |
| LGR GGA-30r-EP | $CO_2$ and $CH_4$ mole fraction | atmosphere/equilibrated air |
| LGR FGGA | $CO_2$ and $CH_4$ mole fraction | atmosphere/equilibrated air |
| Teledyne T200UP | $NO/NO_2/NO_x$ mole fractions | atmosphere |
| Teledyne T200U | $NO/NO_y$ mole fractions | atmosphere |
| Teledyne T400 | $O_3$ mole fraction | atmosphere |
| Teledyne T700U | dynamic dilution calibrator with ozone generator | |
| Teledyne T701H | zero air generator | |
| WS600-UMB | temperature, wind speed, humidity | atmosphere |
| Camsecure, underwater camera | video | ocean |
| 4K HD video camera | video | atmosphere |



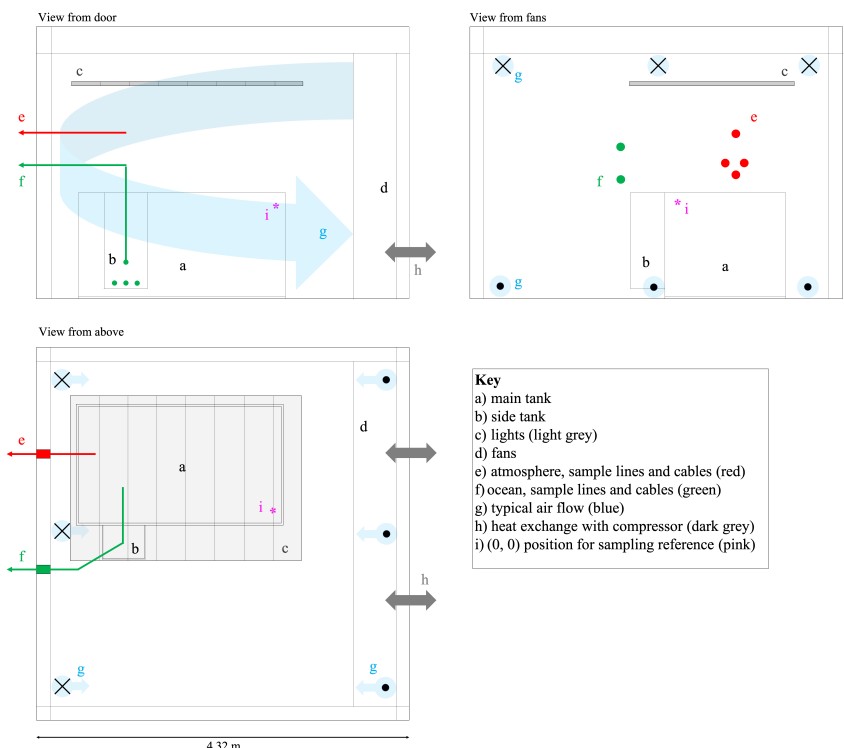

**Figure 3.** To scale schematic diagram of the coldroom. The three panels show orthogonal views from different vantage points. Crosses and dots indicate air flow away from and towards the viewer, respectively. The lights, shown in grey, are made up of eight sets of visible, UV-A, and UV-B triplets.

### 2.2.2 Sea ice

We use chains of digital thermometers to measure temperature, $\theta$, profiles through the ocean and sea ice. These have a resolution of $\frac{1}{16}$ °C and are calibrated against $\theta$ measured by the Seabird 37SIP before each run while they sit in well-mixed water. The chains have 1 cm to 8 cm resolution and are 10 cm to 80 cm long. Wireharps (Notz et al., 2005; Notz and Worster, 2008) are used to measure a liquid fraction profile, $\phi$, through the sea ice using

$$\phi = \frac{\gamma_0 R_0}{\gamma_t(\theta, S_{\mathrm{br}}) R_t}, \tag{1}$$

where $R$ is the resistance between a given wirepair, $\gamma$ is the conductivity of the solution between the wirepairs, and the subscripts $0$ and $t$ indicate the point in time at which the sea-ice front passes a wirepair and some later point in time. We calculate the bulk sea-ice salinity, $S_{\mathrm{si}}$ (g kg$^{-1}$), using

$$S_{\mathrm{si}} = \phi S_{\mathrm{br}}(\theta), \tag{2}$$





where $S_{br}(\theta)$ is a the brine salinity. $S_{br}(\theta)$ is retrieved from the third order polynomial presented in Vancoppenolle et al. (2019) (for natural seawater composition)

$$S_{br}/(g\ kg^{-1}) = -18.7(\theta/^{o}C) - 0.519(\theta/^{o}C)^{2} - 0.00535(\theta/^{o}C)^{3} \tag{3}$$

or Rees Jones and Worster (2014) (for NaCl, fit to the data of Weast (1971)),

$$S_{br}/(g\ kg^{-1}) = -17.6(\theta/^{o}C) - 0.389(\theta/^{o}C)^{2} - 0.00362(\theta/^{o}C)^{3}. \tag{4}$$

Our version of the wireharps use two alternating current frequencies, 2 kHz (as used in Notz et al. (2005)) and 16 kHz. When using the 2 kHz channel we calculated $\frac{\gamma_0}{\gamma_t(\theta,S_{br})}$ following Notz et al. (2005). When using the 16 kHz channel we presume that the electronic double layers driving the conductivity changes are not present and so take $\frac{\gamma_0}{\gamma_t(\theta,S_{br})} = 1$.

We have also deployed fibre optics and photodiode light sensors (Hof, 2019, Table 1) in the sea ice and ocean. Fibre optics can be connected to a spectrometer outside of the coldroom (Ocean optics USB2000+). The light sensors consist of photodiodes,
waterproofed in resin, measuring in the blue, green, red, and clear. The wavelength range integrated by the photodiodes is, to a good approximation, 400 nm to 700 nm. The response of the photodiodes is negligible to light with wavelengths shorter than 400 nm. At wavelengths above 700 nm, the response is generally less than 10 % of the maximum response. Diffusing glass plates sit atop the photodiodes to increase the angle of incidence at which light reaches the diodes. The data sheet for the photodiodes is given in the Supplementary Information.

### 2.2.3 Atmosphere

We use a weather station (WS600-UMB) to measure the temperature, wind speed, and relative humidity of our atmosphere. Two Los Gatos Research (LGR) greenhouse gas analysers measure $CO_2$, $CH_4$, and $H_2O$ mole fractions in the atmosphere and can also analyse the air stream of the Liquicel equilibrator (Table 1). A Teledyne T200UP measures NO and $NO_x$ and a T200U measures NO and $NO_y$. A Teledyne T400 measures ozone. There is also a zero air generator (T701 H, Teledyne) and a T700U
dynamic dilution calibrator with ozone generator.

## 2.3 Experimental protocols

Protocols vary widely between experiments. Here, we provide a typical protocol to help future users visualise the facility and plan experiments.

### 2.3.1 Set-up phase

We set up instrumentation in a dry, clean tank. Sea-ice instrumentation is attached to poles that are free to rise in the vertical as sea ice grows and floats. Ocean and atmosphere instrumentation and sample lines are mounted in a fixed position. The nature of the experiment will determine the state of the tank sides and base. For optical experiments (Section 3.2.1) the tank's inner surfaces should be covered to simplify the light field. We use mirrored sides and a matte black base to approximate infinite lateral boundary conditions and a non-reflective ocean (Figure 1).





To fill the tank, we add 100 kg of salt and mix this with tap or 18.2 MΩ water. The salt can be pure NaCl (used for Sections 3.1.2 and 3.2.1) or a natural sea salt composition (Tropic Marin sea salt classic, used for Section 3.2.2). The tank takes around 18 hours to fill with de-ionised water or a few hours from the tap, and 100 kg of salt dissolves in a few hours under vigorous circulation by the pumps. Alternatively, real sea water can be delivered to the tank. Filling the tank to 1 m gives an ocean volume of around 3.2 m$^3$. Once the tank is full we begin to cool the water. The tank surface can be covered at this stage to reduce

evaporation. With an atmospheric temperature of -20 $^o$C, and an uncovered tank, the temperature of 3.2 m$^3$ of ocean drops by around 1 $^o$C every 4 hours. If the experiment requires an isolated coldroom atmosphere, the coldroom is sealed at this stage. Sea-ice growth can be initialised by continuing to cool under constant temperature and vigorous circulation. Alternatively, once the water is within a few tenths of a degree of its freezing point, turning the pumps to a low setting, or off, initialises sea-ice growth within a few minutes. As a rule of thumb, it takes one full week to go from a dry tank to first sea ice.

**2.3.2    Growth phase**

The water circulation rate and the atmospheric temperature determine the nature of the sea ice (Naumann et al., 2012). Nilas will grow in quiescent water. In circulating water, a layer of grease ice will form that subsequently consolidates. When the growth temperature is less than -25 $^o$C, frost flowers form on the sea-ice surface (Figure 4). By running the heating pads between the main tank and insulation, we are able to maintain free-floating sea ice to 20 cm thickness with no visible water gap

at the tank sides. We have verified that the sea ice is free-floating in several experimental runs by gently pressing the corner of the sea ice and noting that it bobs. With no, or insufficient, side heating, the sea ice attaches to the tank sides and the surface floods (e.g. Rysgaard et al., 2014), resulting in a shiny, liquid surface layer. With insufficient side heating and insulation the ocean tends to supercool (Figure 5). When the ocean supercools, ice grows on the tank sides and instruments, causing problems with measurements and potentially damaging equipment. Growth phases typically last for a few days to a few weeks.

**2.3.3    Sampling protocols**

Underlying water may be sampled throughout, providing the volume is replaced. We sample sea ice either by taking cores using a 7.5 cm diameter Kovacs ice corer or using the procedures outlined by Cottier et al. (1999) to extract sea-ice 'slabs'. Sea-ice cores have a known bias, particularly problematic in young sea ice, as brine is lost from the permeable sea ice near the ocean interface. The method of Cottier et al. (1999) seeks to minimise this bias by collecting sea-ice samples that are still floating,

then freezing the sample and surrounding ocean at -40 $^o$C to immobilise the brine before processing the sample. The method of Cottier et al. (1999) becomes difficult and time consuming when sea ice is thicker than around 25 cm due to limitations of the power tools used to extract the slabs and the weight of the slabs. We cut cores and slabs into discrete vertical profiles using a bandsaw with a pre-cleaned blade at -25 $^o$C.





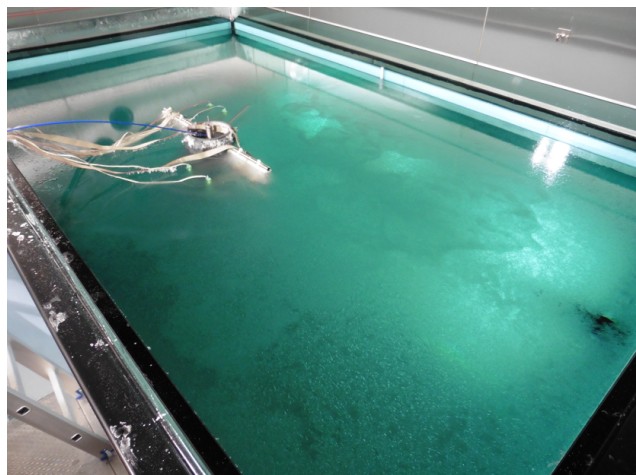

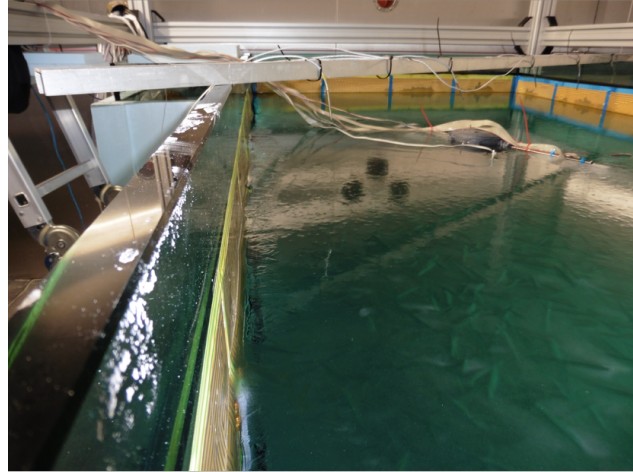

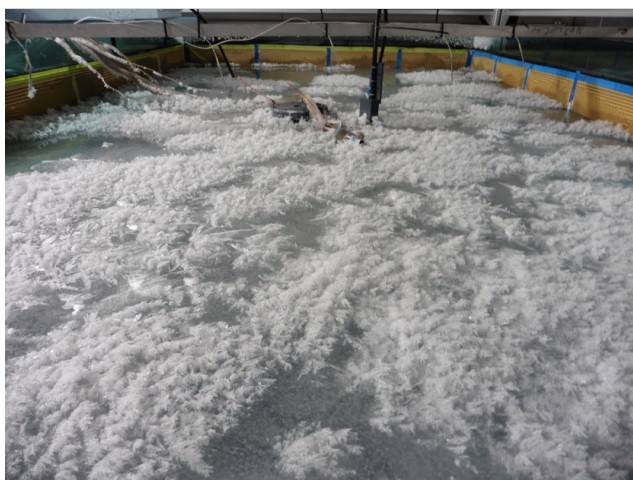

**Figure 4.** Grease ice (top) grown under pumping, nilas (middle) grown in quiescent conditions, and a frost flower field (bottom) grown in the RvG-ASIC.





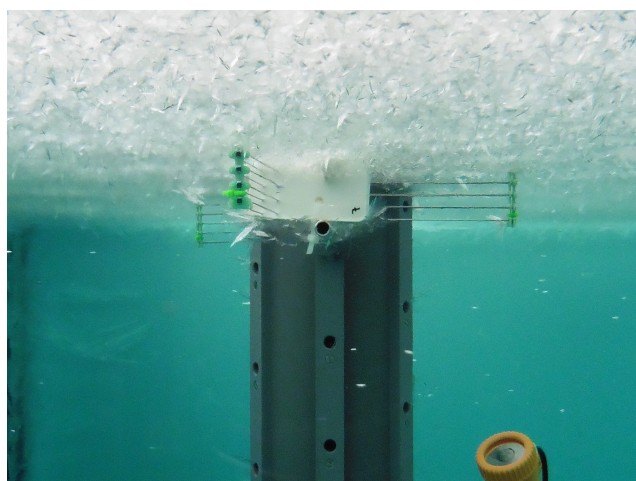

**Figure 5.** A view of a supercooled ocean. Frazil ice crystals are floating upwards in the water column (white flecks) and have nucleated on wireharps well ahead of the advancing sea-ice/ocean interface.

## 3 Characterisation of experimental system

We now turn to a characterisation of our experimental system. First, in Section 3.1, we quantify several parameters regarding the technical capabilities of the facility: the exchange rate between the chamber and the outside, the mixing rate of the water in the tank, and the variability in the light field. In doing so, we aim at providing valuable information to help plan and interpret future studies. Next, in Section 3.2, we present measurements of our experimental sea ice to quantify the extinction of PAR ($\lambda = 400$ nm to $700$ nm), the bulk salinity, and the growth rate of our experimental sea ice. The goal of this section is to investigate how similar our laboratory grown sea ice is to natural sea ice with respect to these important characteristics.

### 3.1 Technical characterisation

#### 3.1.1 Quantifying the coldroom air exchange rate

For most experiments, it is desirable to have the sea-ice/atmosphere interface exposed to the bulk air within the coldroom. Leaving the atmosphere of the tank uncapped ensures that the temperature of the atmosphere overlying the sea ice is responsive to the coldroom atmosphere. When the atmosphere is contained by some headspace the temperature tends to be much warmer than the coldroom (e.g. Loose et al., 2011). Here, we quantify the degree to which the coldroom can be sealed from the outside by deriving the air exchange rate coefficient, $k$, and time constant, $\tau_1$, of $CH_4$ exchanging between the coldroom to the outside (Figure 6).

To do so, we sealed the coldroom and left the tank dry. We then diluted the chamber air by filling it with $N_2$ gas. We performed two dilutions, the first at around 18:00 on 12/02/19 and the other at around 16:00 on 13/02/19. The $CH_4$ concentration in the chamber, $C_{cham}$, was monitored using an LGR (GGA type, Table 1), where measured air was returned to the coldroom in a

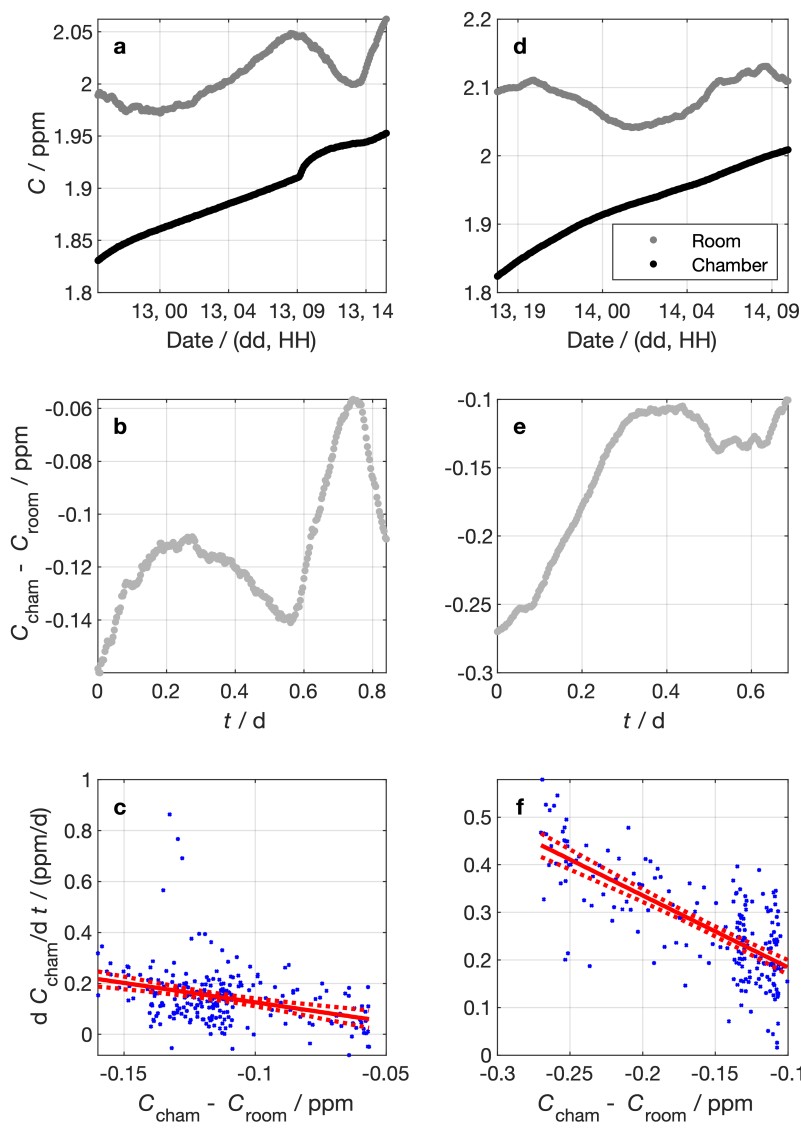

**Figure 6.** Data from experiments to determine the chamber air exchange rate. Panels a to c and d to f show data from the first and second experiments, respectively. The top panels (a and d) show the measured $CH_4$ concentrations in the chamber and the lab outside. Panels b and e show the concentration difference between the chamber and the lab outside. Panels c and f show the linear regressions constructed using Equation 5. Blue dots show individual data points. The red lines show the best fit and 95 % confidence intervals of the linear regression of $\frac{dC_{cham}}{dt}$ against $C_{cham} - C_{room}$. The gradient of the linear regression corresponds to the air exchange rate coefficient, $-k$. For the first experiment $k = (1.5 \pm 0.3)$ d$^{-1}$ and for the second experiment $k = (1.5 \pm 0.1)$ d$^{-1}$.





closed loop (Figure 6, panels a and d). The dilution reduced $C_{\text{cham}}$ by around 0.1 ppm in both dilution periods. We stopped the dilution by stopping the $N_2$ flow, at which point the chamber was sealed to the best of our ability. We measured the $CH_4$ concentration in the outer laboratory, $C_{\text{room}}$, throughout using an LGR (FGGA type, Table 1), and used the difference between

$C_{\text{cham}}$ and $C_{\text{room}}$ as the concentration gradient between the chamber and the outside (Figure 6, panels b and e). Before the experiment, the two LGRs measured the same air for 30 hours. The offset between them was $(6.4 \pm 0.2)$ ppb. This offset was added to $C_{\text{room}}$ values to make the measurements from the two LGRs consistent. Data were averaged in 5 minute bins. We modelled the exchange rate as a first order process,

$$\frac{\text{d}C_{\text{cham}}}{\text{d}t} = -k(C_{\text{cham}} - C_{\text{room}}). \tag{5}$$

We approximate $\frac{\text{d}C_{\text{cham}}}{\text{d}t}$ for each measurement interval using the difference in $C_{\text{cham}}$ between adjacent data points over the five minutes between those data. Panels c (experiment 1) and f (experiment 2) of Figure 6 show the relationship between $\frac{\text{d}C_{\text{cham}}}{\text{d}t}$ and $C_{\text{cham}} - C_{\text{room}}$, and the linear fit. We derived $k = (1.5 \pm 0.3)\,\text{d}^{-1}$ and $k = (1.5 \pm 0.1)\,\text{d}^{-1}$ for the first and second experiments, respectively. Taking the mean $k$ of the two experiments, the air exchange time constant is then given by

$$\tau_l = 1/k = (0.66 \pm 0.07)\,\text{d}. \tag{6}$$

The physical interpretation of this $\tau_l$ is that the concentration difference between the sealed chamber and the outside will reduce by a factor of e every $(0.66 \pm 0.07)$ days.

### 3.1.2   Quantifying the tank mixing rate

For some experiments, it may be necessary to spike the ocean with a chemical. We may, in such cases, want to know when the water will be well mixed with respect to our spike. Similar to the air exchange rate (Section 3.1.1), the degree to which the

chemical has mixed can be quantified using a time constant, $\tau_m$. To derive this time constant, we spiked our tank with saturated NaCl solution by injecting it near the tank base and observed the conductivity of our ocean over time as it mixed under the action of three pumps (Figure 7). The temperature of the ocean was stable to within 0.02 ºC for the hour time period over which we monitored both spikes, so we expect the salinity to be linearly related to the conductivity. The conductivity of the ocean, $\gamma_o$, can then be described by

$$\gamma_o(t) = \text{e}^{-\frac{t}{\tau_m}}(\gamma_o(t_0) - \gamma_{o,\infty}) + \gamma_{o,\infty}, \tag{7}$$

where $\gamma_{o,\infty}$ is the maximum ocean conductivity obtained after the spike and $t_0$ is the time of the spike. Fitting an exponential curve to a manually-defined spike period for the two runs produces $\tau_m = (4.2 \pm 0.1)$ min (run 1) and $\tau_m = (4.1 \pm 0.1)$ min (run 2). We take the mean of the individual experiments to characterise our mixing rate, such that $\tau_m = (4.2 \pm 0.1)$ min. The tank should therefore mix to more than 99% of the perfectly mixed concentration in less than 30 minutes.

### 3.1.3   Quantifying light-field variability

Producing a homogeneous light field in a laboratory environment can be difficult because shading and reflection can cause heterogeneities. The intensity of our lights is also temperature dependent, which is particularly important in the RvG-ASIC



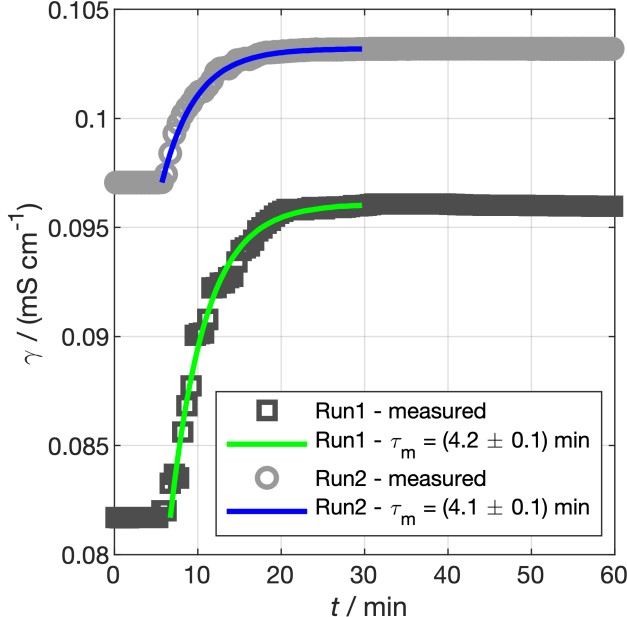

**Figure 7.** Experiments to determine the mixing time constant of our tank, $\tau_\mathrm{m}$. Saturated NaCl solution was spiked into the tank and mixed under pumping. The conductivity of the bulk water is shown by the grey circles. The coloured lines show the prediction of Equation 7 using $\tau_\mathrm{m}$ fit to the data. In reality, the spike in run 2 was a few hours after run 1, but the times for the two spikes have been matched for ease of comparison.

given the wide range of experimental temperatures. In this section, we aim to characterise our light field, both in terms of lateral heterogeneity and with changes in temperature.

To asses the spatial variability, we measured PAR (photosynthetically active radiation, $\lambda = 400$ nm to $700$ nm) across the tank area at 1 m height (Figure 8). These measurements were made using a LiCOR LI-1800 spectroradiometer integrated from 400 nm to 700 nm. The PAR irradiance, $I_\mathrm{PAR}$, at 1 m above the tank base and 25 °C was generally within 80 % of the maximum recorded intensity, $I_\mathrm{PAR,\,max}$, but dropped as low as 60 % in one corner of the tank.

    We measured spectra for UV (UV-A lighting, 320 nm to 380 nm) and PAR (LED lighting, 400 nm to 700 nm) at -30 °C,
-20 °C, -10 °C, and 0 °C using a USB2000+ spectrometer referenced against the Metcon spectral radiometer (Figure 9). To quantify the temperature sensitivity, we integrated the UV and PAR spectral irradiance over their respective wavelength ranges (320 nm to 380 nm and 400 nm to 700 nm, respectively). The maximum irradiance was 5.6 W m$^{-2}$ for UV and 26.4 W m$^{-2}$ for PAR, both at 0 °C. The temperature sensitivity of the irradiance was $(0.16\pm0.01)$ W m$^{-2}$ °C$^{-1}$ for UV and $(0.028\pm0.003)$ W m$^{-2}$ °C$^{-1}$ for PAR. The UV-A lights are more strongly temperature dependent, with irradiance dropping by 83 % between
0 °C and -30 °C, while PAR irradiance dropped by just 3 %.





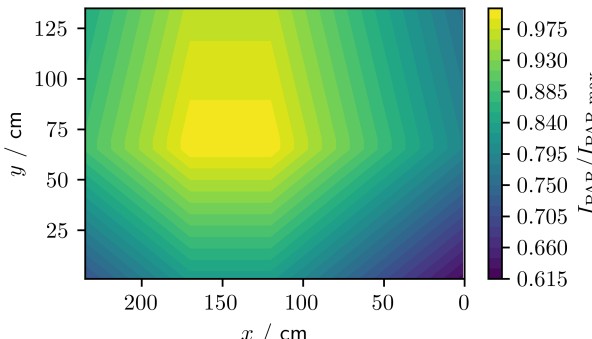

**Figure 8.** Normalised PAR ($\lambda = 400$ nm to 700 nm) irradiance across the footprint of the tank at 1 m height. The (0, 0) cm coordinate corresponds to the pink asterisk in Figure 3.

## 3.2 Sea-ice characterisation

### 3.2.1 Determining PAR extinction coefficients in sea ice

The importance of understanding the sea-ice light field is critical as thinner, fresher, and more transient sea ice becomes more common. Even thin sea ice without snow cover impacts the transmission of PAR to the ocean, and has been seen to
accumulate algal biomass rapidly within the sea ice (Taskjelle et al., 2016). Photochemical production rates within sea ice are also dependent upon sea-ice optical properties. Irradiation through sea ice has been postulated as a source of OH radicals (King et al., 2005) and as a stressor of diatoms that may lead to iodine production from algae (Küpper et al., 2008).

The rationale for UV-Vis illumination experiments in a controlled sea-ice facility is 2-fold: first, to allow experiments investigating the optical properties of sea ice (and potentially other mediums such as snow); and second, to allow simple experimental
simulations of photochemistry/biology occurring in the sea ice, atmosphere, or ocean. The extinction coefficient, $\kappa$, of PAR in sea ice quantifies the rate of attenuation of PAR with depth. Here, we present measurements of $\kappa$ for sea ice grown in the RvG-ASIC and compare those to previously reported values.

To quantify $\kappa$ in our experimental sea ice, we performed an experiment using visible lighting, measuring a vertical profile of irradiance using eight photodiodes (Section 2.2.2). The photodiodes respond dominantly to 400 nm to 700 nm light, and
our lighting provides irradiance predominantly in the 400 nm to 700 nm range, so $\kappa$ as measured by the photodiodes is a good estimate of $\kappa$ for PAR. The tank sides were covered with mirrored tape to approximate infinite lateral boundary conditions, and the tank base was covered with a matte black plastic sheet to approximate no reflection from the ocean. A 3 mm opal polycarbonate sheet was placed 20 cm below the light rack to create a diffuse light field, and black curtains were draped between the lights and the tank to prevent reflection off the shiny coldroom walls (Figure 1). We deployed the photodiodes
with 2.8 cm vertical resolution. The sensors were calibrated pre-deployment against the met-con spectral radiometer. Before each experimental run, while sitting in open water, an additional correction was applied to the *in situ* sensor output to bring it



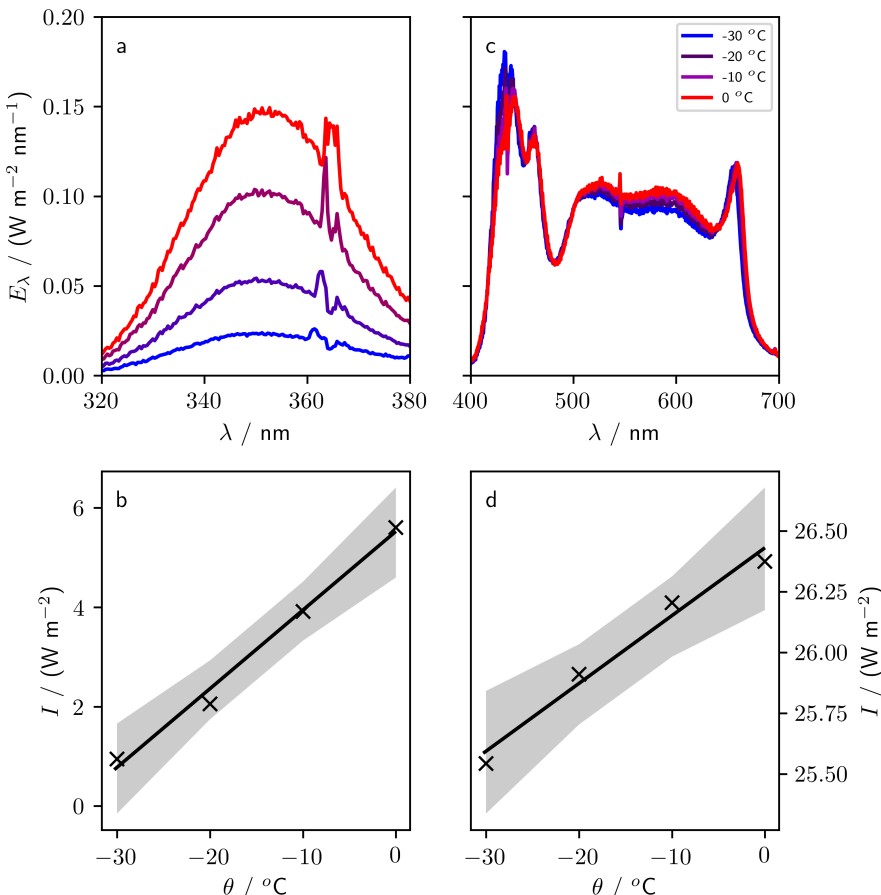

**Figure 9.** Sensitivity of lights to variations in temperature. Panels a (UV) and c (PAR) show spectra taken at four temperatures between 0 °C and -30 °C. Panels b (UV, 320 nm to 380 nm) and d (PAR, 400 nm to 700 nm) show the irradiance, $I$ (integrated spectral irradiance over the respective wavelength ranges), at each temperature, $\theta$. The gradient of the regression of $I$ against $\theta$ is $(0.028 \pm 0.003)$ W m$^{-2}$ °C$^{-1}$ for PAR and $(0.16 \pm 0.01)$ W m$^{-2}$ °C$^{-1}$ for UV.





into line with depth profiles measured using the spectral radiometer. We performed four experiments at growth temperatures of -10 °C, -20 °C (two runs), and -30 °C (Table 2). For each run, we present vertical light profiles taken at the end of the growth phase, as well as $\kappa$ calculated by two methods (Figure 10 and Table 3). Method A (Ehn et al., 2004; Kauko et al., 2017) uses

$$\kappa = \ln\left( (1 - \zeta_s) \frac{I(+1.4 \text{ cm})}{I(z)} \right) / z. \tag{8}$$

The depth, $z$, is chosen to be the shallowest sensor not yet frozen into the sea ice, and spectral reflectance, $\zeta_s$, assumed to be 0.05. Ehn et al. (2004) measured $\kappa$ ranging from 3.1 m$^{-1}$ to 4.7 m$^{-1}$ in 24 cm to 28 cm-thick sea ice (Table 3). Kauko et al. (2017) measured $\kappa$ ranging from 2.9 m$^{-1}$ to 4.7 m$^{-1}$ in 17 cm to 27 cm-thick sea ice. For method B, we also calculated $\kappa$ by fitting a linear model to

$$\ln(I/(\text{W m}^{-2})) = -\kappa z + c, \tag{9}$$

similar to (Marks et al., 2017). Equation 9 represents a light field decaying exponentially with depth. While method A and B both provide estimates of $\kappa$, they do so in different ways, with method A using measurements external to the sea ice while method B uses only measurements within the sea ice. Previous studies, compared to our measurements in Table 3, have calculated $\kappa$ by iteratively solving the Dunkle and Bevans (1956) photometric model such that $\kappa$ provides the best fit to measured albedo and transmission (Perovich and Grenfell, 1981) — which we call method C.

Measured light profiles taken at the end of four experiments are shown in Figure 10. The thickness varied by up to 3 cm between the four experiments. At the end of the experiments, runs 1 and 2 had six sensors frozen into the sea ice and experiments 3 and 4 had five sensors frozen into the sea ice. The light intensity at 1.4 cm above the sea ice was between 7.7 W m$^{-2}$ and 8.9 W m$^{-2}$ and reduced to between 2.8 W m$^{-2}$ and 3.6 W m$^{-2}$ at 18.2 cm depth. For experiments 1 and 4, the shallowest frozen sensor measured higher intensity than the sensor above the sea ice, a phenomenon that is predicted by Jiang et al. (2005) and that is due to the change in refractive index across the air-ice interface.

Using method A, our calculated $\kappa$ ranges from 3.7 m$^{-1}$ (experiment 4) to 6.1 m$^{-1}$ (experiment 2). Using method B, our calculated $\kappa$ ranges from 4.4 m$^{-1}$ (experiment 2) to 6.2 m$^{-1}$ (experiment 4). For both methods, the range of $\kappa$ observed in our tank overlaps with the range of $\kappa$ observed previously for thin sea ice. These results build confidence that the RvG-ASIC is a useful tool for future biological or photochemical experiments in young sea ice, and validate this particular experimental set up for experiments involving visible light.

### 3.2.2  Estimating sea-ice bulk salinity

Bulk salinity is a sea-ice state variable that, when measured alongside temperature, can allow estimation of the sea-ice liquid fraction. Growing sea ice desalinates rapidly by gravity drainage (Notz and Worster, 2008). The bulk salinity of natural, young sea ice has been measured using cores (Weeks and Lee, 1962) and wireharps (Notz and Worster, 2008). Weeks and Lee (1962) sampled young sea ice in North Star Bay, Greenland, in growing sea ice between 5 cm and 23 cm thick. Salinities ranged from around 5 g kg$^{-1}$ to 20 g kg$^{-1}$ vertically with mean salinities decreasing from 20 g kg$^{-1}$ early in the growth to around 10 g kg$^{-1}$ after 10 days. Notz and Worster (2008) cut a hole in Arctic sea ice and measured the salinity profile, *in situ*, as it re-froze





**Table 2.** Overview of experiments used to calculate $\kappa$. $\theta_{atm}$ gives the mean atmospheric temperature during the experiment. $\frac{\psi}{\psi_{sat}}$ is the mean relative humidity during the experiment, with $\psi$ representing humidity and the subscript sat indicating saturation. $t_f$ gives the duration of the freeze up period. $h_{si}$ gives the maximum thickness attained, at which point $\kappa$ was measured.

| Experiment | $\theta_{atm}$ / °C | $\frac{\psi}{\psi_{sat}}$ | $t_f$ / d | $h_{si}$ / cm | Surface characteristics |
|---|---|---|---|---|---|
| 1 | $-18.9 \pm 0.4$ | $0.59 \pm 0.05$ | 6.1 | 18 | Flooded |
| 2 | $-26.6 \pm 0.9$ | $0.59 \pm 0.03$ | 5.1 | 18 | Frost flowers |
| 3 | $-9.2 \pm 0.9$ | $0.52 \pm 0.03$ | 10.1 | 15 | |
| 4 | $-18.0 \pm 0.5$ | $0.60 \pm 0.05$ | 7.0 | 15 | |

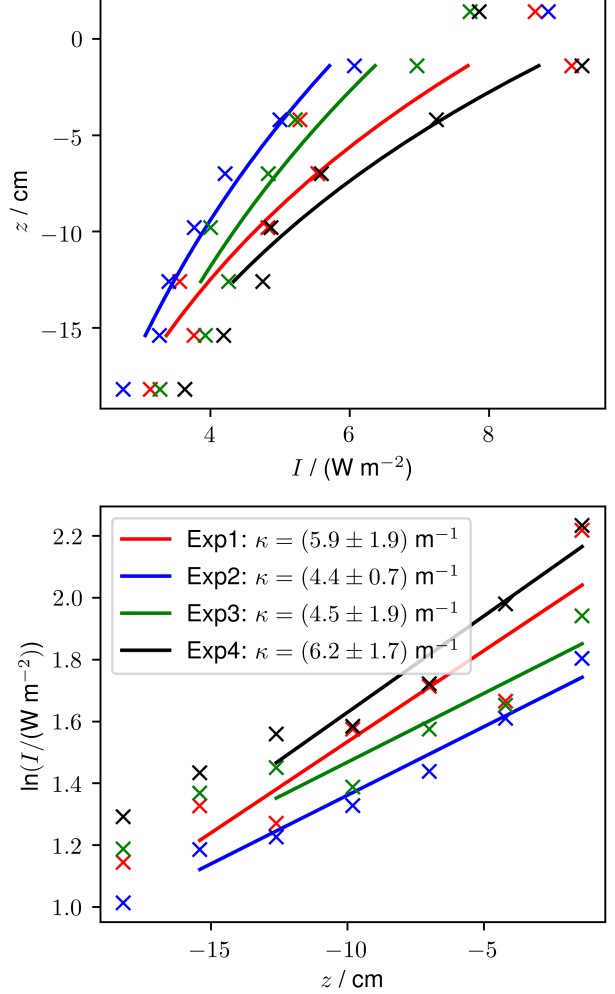

**Figure 10.** Irradiance profiles (a) and linear regressions of $\ln(I / \mathrm{W\ m^{-2}})$ vs $z$ (b) used to produce extinction coefficients (Equations 8 and 9) from the final profile of the freezing period. The legend in panel b gives the extinction coefficient for a given profile with one standard error.

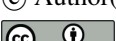



**Table 3.** Comparison of PAR extinction coefficients ($\kappa$, wavelength range 400 nm to 700 nm) produced by this work and by different studies.

| Experiment | $\theta_{atm}$ / °C | $h_{si}$ / cm | $\kappa$ / m$^{-1}$ | | |
|---|---|---|---|---|---|
| | | | A | B | C |
| This work, Exp 1 | $-18.9 \pm 0.4$ | 18 | 5.3 | $5.9 \pm 1.9$ | |
| This work, Exp 2 | $-26.6 \pm 0.9$ | 18 | 6.1 | $4.4 \pm 0.7$ | |
| This work, Exp 3 | $-9.2 \pm 0.9$ | 15 | 4.1 | $4.5 \pm 1.9$ | |
| This work, Exp 4 | $-18.0 \pm 0.5$ | 15 | 3.7 | $6.2 \pm 1.7$ | |
| Nice[1] | -10 to -20 | 17 to 27 | 2.9 to 4.7 | | |
| Santala bay, Finland[2] | <-5 to 5 | 24 to 28 | 3.1 to 4.7 | | |
| Laboratory[3] | -10 | 28 | | | 2.5 |
| Laboratory[3] | -30 | 28 | | | 3 |
| Laboratory[4] | -15 | 40 | | 3 to 10 | |

[1] Kauko et al. (2017); [2] Ehn et al. (2004); [3] Perovich and Grenfell (1981); [4] Marks et al. (2017).

A - Equation 8

B - Equation 9. Values from this work are broadband $\kappa$ while the range of values given for Marks et al.

(2017) cover a range of $\kappa$ at wavelengths between 350 nm and 650 nm.

C - Dunkle and Bevans (1956) model iteratively solved.

up to 17 cm thickness (the depth range of their instrument). Over two experiments, they measured bulk salinities ranging from
around 35 g kg$^{-1}$ at the sea-ice/ocean interface to around 4 g kg$^{-1}$ in the interior. Mean salinities decreased from around 35 g kg$^{-1}$ towards 10 g kg$^{-1}$ (experiment 1) and around 15 g kg$^{-1}$ (experiment 2). Here, we present bulk salinities for sea ice grown in the RvG-ASIC, comparing it to the range of salinities observed in nature.

Quantifying the vertical bulk salinity profile in sea-ice cores – the most common sea-ice sampling methodology – is difficult due to the known bias during coring of brine loss and bulk salinity underestimation. Several other methodologies have been
proposed. Cottier et al. (1999) present a destructive sampling methodology that attempts to prevent brine loss upon sampling (Section 2.3.3). Notz et al. (2005) present instrumentation, called a wireharp, to quantify the *in situ* bulk salinity profile. Several groups have presented gravity drainage parameterisations that quantify the bulk salinity profile during sea-ice growth (Griewank and Notz, 2013; Rees Jones and Worster, 2014; Cox and Weeks, 1988; Vancoppenolle et al., 2010; Turner et al., 2013; Jeffery et al., 2011), noting that gravity drainage is the dominant process redistributing salt in growing sea ice (Notz and
Worster, 2009).

We performed two sea-ice growth experiments in the RvG-ASIC and estimated the sea-ice salinity by: 1) constructing a salt and mass budget, 2) taking sea-ice cores, 3) taking sea-ice slabs (Cottier et al., 1999), 4) using a wireharp (Notz et al., 2005), and 5) using the Griewank and Notz (2013) gravity drainage parameterisation. Our artificial ocean was composed of Tropic Marin sea salt and deionised water such that $S_o = (31.3 \pm 0.1)$ g kg$^{-1}$. We grew free-floating sea ice to $(11.0 \pm 0.9)$
cm (run 1) and $(16.4 \pm 2.7)$ cm (run 2) thickness (mean and standard deviation of measured ice core thickness). Measurements of sea-ice temperature, thickness, and the initial ocean salinity were used to force the model, which has been used previously to model experiments in the RvG-ASIC (Garnett et al., 2019; Thomas et al., 2020). We used the Griewank and Notz (2013)





gravity drainage parametrisation because it performed well in previous studies (Thomas et al., 2020) and has tuning parameters shown to perform well for Arctic field data (Griewank and Notz, 2015). The model has two tuning parameters: the critical
Rayleigh number, $Ra_c$, and the desalination strength, $\alpha$. We estimate an uncertainty on the modelling by running the model for each of the three sets of tuning parameters presented in Griewank and Notz (2015) (Table 4). The model is forced by measured temperature profiles and sea-ice thickness in lieu of modelled thermodynamics (Figure 11). Cores and slabs were extracted according to Section 2.3.3 and sectioned vertically into 1 cm to 2 cm layers. Ice samples were melted and placed in a 20 $^o$C thermostatic bath. Conductivity was measured using a conductivity probe (Orion), calibrated with certified reference
material (Thermo Scientific Eutech Handheld Meters Calibration Solution). Bulk Salinity was derived using the GSW toolbox (McDougall and Barker, 2011). We discarded three discrete samples (one from the cores and two from the slabs for run 1) because of punctured sample bags. A single wireharp profile was taken in the middle of the tank and used to derived $\phi$ and $S_{si}$ (Equations 1 and 2).

The mass balance was constructed by conservation of mass and salt from the start of each run ($t_0$) to the final sampling
($t_{end}$). The conservation of mass gives

$$m_{sys} = m_o + m_{si} \tag{10}$$

where $m$ is mass and the subscripts sys, o, and si indicate the experimental system, ocean, and sea ice compartments, respectively. Noting that $m_{sys} = m_o(t_0)$, the conservation of salt at the end of the experiment ($T_{end}$) gives

$$m_{sys}S_o(t_0) = m_o S_o(t_{end}) + m_{si}\bar{S}_{si}. \tag{11}$$

Our desired variable is the vertically integrated bulk sea-ice salinity, $\bar{S}_{si}$, which we recover by substituting Equation 10 into Equation 11, giving

$$\bar{S}_{si} = \Phi \Delta S + S_o(t_{end}) \tag{12}$$

where

$$\Delta S = S_o(t_0) - S_o(t_{end}) \tag{13}$$

and $\Phi$ gives the ratio of the mass of the experimental system to the mass of the sea ice at final sampling, such that

$$\Phi = \frac{h_o(t_0)\rho_o(t_0)}{h_{si}\rho_{si}}. \tag{14}$$

The error on Equation 12 was calculated by Gaussian propagation, such that

$$\sigma(\bar{S}_{si}) = \sqrt{\Phi^2 \Delta S^2 \left( \left( \frac{\sigma(\Phi)}{\Phi} \right)^2 + \left( \frac{\sigma(\Delta S)}{\Delta S} \right)^2 \right) + \sigma(S_o)^2}, \tag{15}$$

where $\sigma$ gives the uncertainty of the individual terms. The dominant uncertainty, accounting for $> 95\%$ of $\sigma(\bar{S}_{si})$, propagates
from $\sigma(\Phi)$ and derives chiefly from the variability in measured $h_{si}$. The errors on $S_o(t_0)$ and $S_o(t_{end})$ are highly correlated





given the stability of the salinity sensor over the short duration of the experiment, and as such $\sigma(\Delta S) << 0.01$ g kg$^{-1}$ and has a negligible impact on $\sigma(\bar{S}_{\mathrm{si}})$. The sensitivity of $\bar{S}_{\mathrm{si}}$ to our choice of $\rho_{\mathrm{si}}$ is around 0.02 (g kg$^{-1}$) / (kg m$^{-3}$), such that a 20 kg m$^{-3}$ increase in $\rho_{\mathrm{si}}$ causes a 0.4 g kg$^{-1}$ increase in our estimated $\bar{S}_{\mathrm{si}}$, well within our uncertainty bounds.

We first turn to the vertical profiles of $S_{\mathrm{si}}$ (Figure 11). For both runs, the wireharp profile at low frequency including the electrical double-layer correction proposed by Notz et al. (2005) is very similar to the high-frequency wireharp profile without such correction. This confirms that at higher frequencies the impact of the double layer becomes less important and no more correction to the profile is necessary. For simplicity, and because the high frequency channel has not yet registered ice growth in the deepest run 1 sensor, we proceed by discussing only the low frequency channel. For both runs, and for all profiles, there is a salinity maximum near the sea-ice/ocean interface ($z = h_{\mathrm{si}}$). The model and the wireharps converge towards $S_{\mathrm{o}}$ as $z$

approaches $h_{\mathrm{si}}$. The cores and the slabs produce the lowest $S_{\mathrm{si}}$ near this lower interface, with the slabs giving higher $S_{\mathrm{si}}$ relative to the cores in this region in run 2. For run 1, the wireharps generally measure the highest $S_{\mathrm{si}}$, the cores and slabs are similar and measure the lowest $S_{\mathrm{si}}$, and the model is intermediate. For run 2, the cores, slabs, wireharps, and model are generally consistent between $z/h_{\mathrm{si}} = 0.6$ to $0.1$. Near the upper interface, the model predicts higher $S_{\mathrm{si}}$ than the measurements, and the wireharps give the lowest $S_{\mathrm{si}}$. For run 1, the cores and slabs are consistent throughout, but the wireharps, model, and discrete

samples generally disagree. Towards the sea-ice/atmosphere interface ($z = 0$), the wireharps produce minima in $S_{\mathrm{si}}$. The cores and slabs show an $\approx 1$ g kg$^{-1}$ increase in $S_{\mathrm{si}}$ for the shallowest layer relative to the layer below, forming a 'C'-shaped profile. Modelled $S_{\mathrm{si}}$ at $z = 0$ is around 10 g kg$^{-1}$ greater than in the interior, forming a 'C'-shaped profile that is more pronounced than in the discrete samples at both interfaces. The difference in $\theta$ between the wireharps and the forcing is at most 0.2 ºC which, depending on $S_{\mathrm{br}}$, translates to a 2 % to 3 % difference in $S_{\mathrm{si}}$.

The mass balance produces $\bar{S}_{\mathrm{si}} = (11.0 \pm 2.0)$ g kg$^{-1}$ (run 1) and $\bar{S}_{\mathrm{si}} = (11.0 \pm 4.1)$ g kg$^{-1}$ (run 2) (mean and standard deviation, Table 4). These values represent the vertically integrated bulk sea-ice salinity, averaged over the tank footprint. We compare the measurements and model to the mass balance by calculating $\bar{S}_{\mathrm{si}}$ for the profile produced by each method at the end of each run. For the cores and slabs, we first averaged the individual layer measurements for each sample, then took the average of these samples to be $\bar{S}_{\mathrm{si}}$. For the wireharps, we linearly interpolated measured $S_{\mathrm{si}}$ to the mid-point between each measurement,

and took the mean of these interpolated $S_{\mathrm{si}}$ to produce $\bar{S}_{\mathrm{si}}$. In this way, we more closely approximate the treatment of the cores and slabs, for which the edge of the bottommost layer is at $z = h_{\mathrm{si}}$. Model $\bar{S}_{\mathrm{si}}$ is the mean of the $S_{\mathrm{si}}$ of individual model layers. The cores and slabs perform similarly, underestimating the mass balance for each run. The wireharps overestimate the mass balance in run 1 and capture the run 2 mass balance to within $1\sigma$. The best estimate model $\bar{S}_{\mathrm{si}}$ is consistent with the mass balance for both runs.

The underestimation of $\bar{S}_{\mathrm{si}}$ by the cores (Table 4) highlights a known bias in sea-ice core bulk salinity measurement, that of brine loss upon sampling. This bias is apparent in the vertical profiles (Figure 11), where cores give the lowest estimates of $S_{\mathrm{si}}$ in the lower third of the profile, which we expect to be most affected by brine loss. The slabs perform similarly to the cores, potentially due to brine drainage during the shock freezing process. The slabs retain more brine than the cores in run 2, as shown by their higher $S_{\mathrm{si}}$ in the lower portion and their higher $\bar{S}_{\mathrm{si}}$. Previous work has tuned gravity drainage parameterisations

to slab $S_{\mathrm{si}}$ measurements (Thomas et al., 2020), and these tuning parameters may therefore be biased to predict too much





**Table 4.** Results for the mean bulk salinity, $\bar{S}_{si}$, calculated from five methodologies for two experimental runs. $t_f$ is the duration of the freezing period and $h_{si}$ is the mean sea-ice thickness as measured in cores at the end of each run. The number of samples/profiles is given in brackets. With the exception of the model, all uncertainties are $1\sigma$ standard deviations. Mass balance, core, and slab uncertainties were calculated based on repeat measurements. Wireharp uncertainties were calculated by propagation of the uncertainties on $R_0$ and $\theta$ for individual wirepairs. The model gives $\bar{S}_{si}$ as predicted using the three parameter sets presented in Griewank and Notz (2015). Low bound: $Ra_c = 3.23$, $\alpha = 0.000681$ kg m$^{-3}$s$^{-1}$. Best estimate: $Ra_c = 4.89$, $\alpha = 0.000584$ kg m$^{-3}$s$^{-1}$. High bound: $Ra_c = 7.10$, $\alpha = 0.000510$ kg m$^{-3}$s$^{-1}$.

| run | $t_f$ / d | $h_{si}$ / cm | $S_o$ / (g kg$^{-1}$)$^\dagger$ | | $\bar{S}_{si}$ / (g kg$^{-1}$) | | | | |
|---|---|---|---|---|---|---|---|---|---|
| | | | Start | End | Mass balance | Cores | Slabs | Wireharp$^\S$ | Model |
| 1 | 3.6 | $11.0 \pm 0.9$ (10) | 31.3 | 33.9 | $11.0 \pm 2.0$ (10) | $8.2 \pm 0.2$ (3) | $8.0 \pm 0.5$ (2) | $15.8 \pm 0.5$ | 12.9 [12.0 to 13.7] |
| 2 | 6.5 | $16.4 \pm 2.7$ (10) | 31.3 | 35.4 | $11.0 \pm 4.1$ (10) | $6.5 \pm 0.5$ (3) | $7.3 \pm 0.2$ (3) | $10.0 \pm 0.4$ | 11.6 [10.8 to 12.5] |

$^\dagger \pm 0.1$ g kg$^{-1}$

$^\S$ low channel with $\gamma$ correction

desalination. The wireharp performance is more variable, overestimating $\bar{S}_{si}$ in run 1 and capturing the mass balance in run 2. The performance of the model, forced with measured $\theta$ and $h_{si}$, solely reflects the gravity drainage parametrisation and its tuning (Griewank and Notz, 2013, 2015), incorporating minimal error from thermodynamics. Only the model captures the mass balance for both runs. The mass balance is, for both runs, within the range of young sea-ice salinities observed in nature

(Weeks and Lee, 1962; Notz and Worster, 2008)

### 3.2.3 Quantifying sea-ice growth rates

The growth rate of sea ice depends on the balance of fluxes at the sea-ice/ocean interface, the thermal conductivity of the sea ice ($K$), sea-ice thickness $h_{si}$, and the absolute sea-ice surface temperature ($T_s$). Growth rates in young sea ice have been observed to range from 2.7 cm d$^{-1}$ to 12 cm d$^{-1}$ (Wakatsuchi and Ono, 1983). Following Stefan (1889), the change in sea-ice thickness

is

$$\frac{dh_{si}}{dt} = -F_C/(\rho_i L) \tag{16}$$

where $L$ gives the latent heat of fusion of water, and $F_C$ gives the vertical conductive heat flux through the sea ice. $F_C$, was calculated using

$$F_C = K(T_s - T_f)/h_{si}. \tag{17}$$

The vertically-averaged thermal conductivity is $K = K_i(1 - \phi) + K_{br}\phi$, where the subscripts i and br indicate pure ice and brine, respectively. $T_f$ is given by the salinity-dependant freezing point of the ocean, and $T_s$ is found by iteratively solving the surface energy balance of the sea ice

$$F_{L\downarrow} - F_{L\uparrow} + F_S + F_C = 0 \tag{18}$$



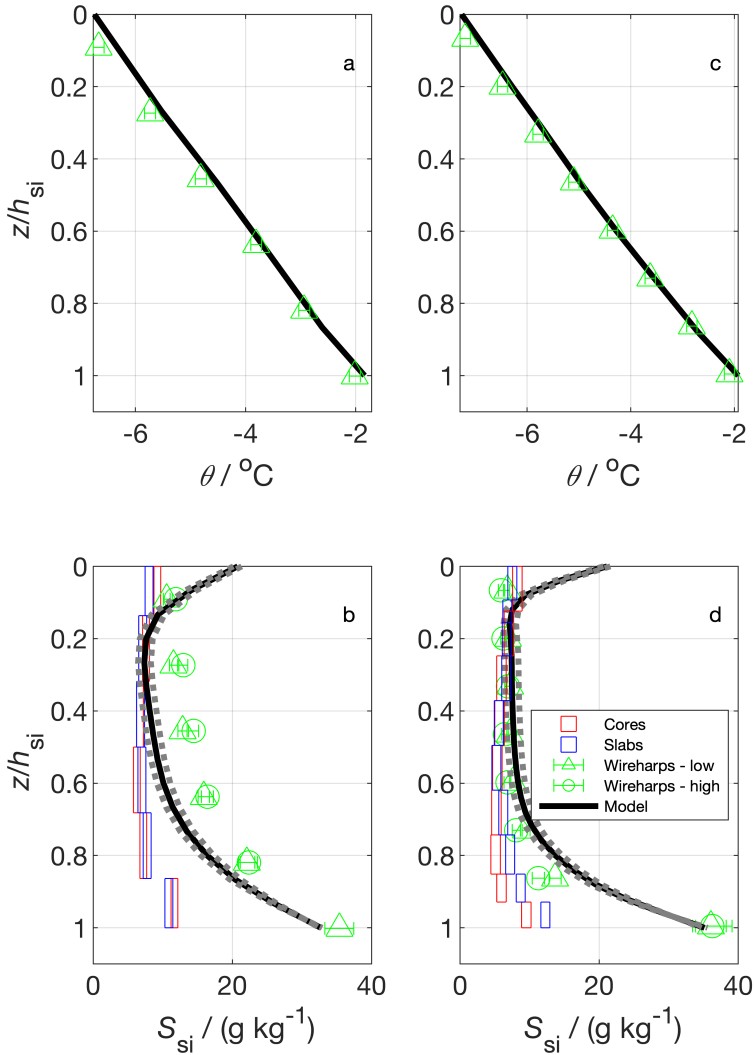

**Figure 11.** Vertical temperature (a and c) and bulk salinity profiles (b and d) for two experimental runs. In a and c the model temperature shows the model forcing, which was produced using measured temperature profiles; b and d show salinity estimated from cores, slabs (Cottier et al., 1999), wireharps (Notz et al., 2005), and a gravity drainage model. The horizontal box length shows the median $1\sigma$ standard deviation from repeat measurements at a given depth (cores and slabs) and the vertical box length shows the depth covered by the sample layer. Wireharp errors were calculated using methodology presented in Zeigermann (2018). The model used the Griewank and Notz (2013) gravity drainage parameterisation with tuning parameters taken from Griewank and Notz (2015). The solid line represents their best estimate tuning parameter set. The dotted bounds on the model show output using two other plausible sets of tuning parameters presented in Griewank and Notz (2015).





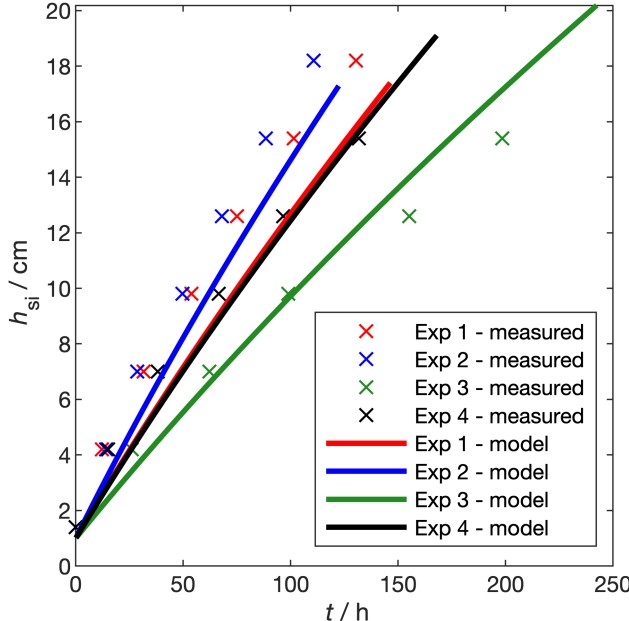

**Figure 12.** Thickness measured from temperature profiles during the experiments presented in Section 3.1.3 and modelled using Equation 16.

where sensible and latent heat fluxes have been neglected because the wind speed in our experiments was less than $0.1$ m s$^{-1}$.

We parameterised the downwelling longwave, $F_{L\downarrow}$, following Efimova (1961), as

$$F_{L\downarrow} = \sigma(0.746 + 0.0066(\psi/\text{mbar}))T_a^4, \qquad (19)$$

where $\sigma$ is the Stefan-Boltzmann constant, $\psi$ is the water vapour pressure, and $T_a$ is the absolute temperature of the atmosphere. The upwelling longwave, $F_{L\uparrow}$, is taken to be

$$F_{L\uparrow} = \sigma\epsilon T_s^4, \qquad (20)$$

where the emissivity of sea ice is $\epsilon = 0.99$. We compare the growth rate of our experimental sea ice to natural sea ice by modelling and measuring the growth rate of the four experimental runs presented in Section 3.2.1 and Table 2 (Figure 12). When the temperature of a sensor at depth $z$ dipped by $0.1$ °C below the water temperature for 30 minutes, we took $h_{si} = z$. For each run, we forced the sea-ice/growth and desalination model (Section 3.2.2, (Thomas et al., 2020)) with the mean $T_a$ and $\psi$ during the freezing period, as measured by the weather station (Table 2). Incoming shortwave, $F_S$, was taken to be 8 W m$^{-2}$

based on measurements of PAR just above the sea-ice surface (Section 3.1.3). In this case, Equations 16 to 20 were used to calculate $T_s$, $h_{si}$, and the internal sea-ice temperature profile. $T_f$ was calculated from the salinity-dependent freezing point of the model ocean.

Over the four experiments, measured growth rates range from 2 cm d$^{-1}$ (experiment 3, warmest air temperature) to 4 cm d$^{-1}$ (experiment 2, coldest air temperature). These growth rates are within the range of those reported by Wakatsuchi and Ono


(1983). Our growth rates are at the low end of this range because the low wind speed in our facility effectively removes the latent and sensible heat fluxes present in the field. The change in $h_{si}$ with time is not linear; rather, as has been observed before (Anderson, 1961), the rate of increase in $h_{si}$ decreases with increasing $h_{si}$. Modelled thickness captures the non-linearity in the measured growth rates and the order of the growth rates, with coldest temperatures producing fastest growth. Modelled thickness deviates from the measurements by up to 3 cm.

Modelling thickness in this way is useful for planning experiments but – considering temperature profiles are measured during each experiment – measuring temperature and thickness gives better precision. Growth rates in the RvG-ASIC are within the range of those measured in the field and are in agreement with thermodynamic modelling (Stefan, 1889).

## 4 Conclusions

We have described the Roland von Glasow Air-Sea-Ice Chamber (RvG-ASIC), the suite of instruments supporting it, and given
an overview of the protocols used to run experiments in the facility. We presented technical results from experiments in the facility showing: 1) the time constant for air exchanging between our sealed chamber and the outside is $(0.66 \pm 0.07)$ days; 2) the time constant for mixing our tank is $(4.2 \pm 0.1)$ minutes; 3) the integrated irradiance of UV-A and PAR at 0 ºC are 5.6 W m$^{-2}$ and 26.4 W m$^{-2}$, respectively; 4) the temperature sensitivity of our LED and UV-A lighting is $(0.028 \pm 0.003)$ W m$^{-2}$ ºC and $(0.16 \pm 0.01)$ W m$^{-2}$ ºC, respectively; and 5) PAR intensity varies by around 10 % near the centre of the tank
but is as low as 60 % near the corners. These technical results can be used to design and interpret future experiments. We also characterised our experimental sea ice showing: 6) the extinction coefficient of PAR in our experimental sea ice is within the range of previously observed PAR extinction coefficients in young sea ice; 7) the bulk salinity of our experimental sea ice is similar to that observed in nature and is in agreement with halo-dynamic modelling; and 8) the growth rate of our experimental sea ice is within the range of previously reported growth rates and is in reasonable agreement with thermodynamic modelling.
This characterisation builds confidence that the RvG-ASIC produces experimental sea ice that is a reasonable analogue of natural sea ice for these important parameters.

The RvG-ASIC is a powerful and versatile tool for studying sea ice and has potential to investigate physics, chemistry, and biology. It is best suited to process studies, bridging the gap between numerical models and reality. The facility was named in honour of its founder, who won funding for the facility, led its design and construction, but sadly died in September 2015,
before it could be put to full use.

*Code and data availability.* All data, plot scripts, and model code used to produce this article are provided as supplementary information accessible at https://doi.org/10.5281/zenodo.4058611 (submission version). Upon final publication a finalised supplementary information will be uploaded to a Zenodo repository, again accesible via a DOI.



*Author contributions.* MT prepared the manuscript with JF, JK, and DN. All authors were involved in some of the experimental work. MT

did the modelling. JF managed the facility from 2015 to 2018 and OC managed the facility from 2018 to 2020, both under the supervision of

JK.

*Competing interests.* The authors have no competing interests.

*Acknowledgements.* Roland von Glasow was instrumental in the design, construction, and scientific vision of the facility. Thanks to Bill Sturges, Dorothee Bakker, Martin Vancoppenolle, and Finlo Cottier for their time and scientific input to the RvG-ASIC. Jeremey Wilkinson

and Martin King provided much useful advice and loaned us equipment. Thanks also to the technical support at UEA: Andy Macdonald, Stuart Rix, Dave Blomfield, Nick Griffin, Gareth Flowerdue, Ben McLeod, and Nick Garrard. This work received funding from the European Research Council under the European Union's Seventh Framework Programme (FP7-2007-2013, Grant Agreement 616938) and the Horizon 2020 research and innovation program through the EUROCHAMP-2020 Infrastructure Activity under Grant Agreement 730997, as well as the University of East Anglia. OT and MTr were supported by an internship granted by the Environmental Sciences department at UEA.



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
