# Peer review of "The Roland von Glasow Air-Sea-Ice Chamber (RvG-ASIC): an experimental facility for studying ocean/sea-ice/atmosphere interactions"

_Atmospheric Measurement Techniques, 2020_

## Referee Comment (RC1) · Anonymous Referee #1 · 20 Nov 2020

Review of manuscript entitled "The Roland von Glasow Air-Sea-Ice Chamber (RvG-ASIC): an experimental facility for studying ocean/sea-ice/atmosphere interactions by M Thomas et al.

This manuscript describes a state-of-the-art laboratory facility for preparing laboratory-grown sea ice in a setting that can be exploited for process study. The manuscript should be of high interest to the scientific community. The text is well written, concise, accurate, and the figures are appropriate. I have no concerns about this manuscript and recommend it be published almost as is.

I have only a few, very minor comments and a few questions for the authors: 105:

[Figure]

"carrying out measurements" instead of "measuring"? Fig.2 impossible to distinguish line shades / colors 413, 414: "air temperatures" are "high" or "low", not "warm" or "cold"

What is the thickest ice that can be grown in this facility? The text says it can be 20cm and still be floating. It's not clear whether the ice can be grown thicker?

How is ice growth prevented in the side tank? I assume the side tank is insulated on top, but it's not stated explicitly.

When sea ice grows, brine rejection at the growing interface necessarily increases the salinity of the ocean, or in this case, in the tank water. I wonder if the side tank in this laboratory setup could be used to help ameliorate this shortcoming associated with a finite-depth tank?

---

## Referee Comment (RC2) · Brice Loose (Referee) · 29 Nov 2020

Overview: The manuscript "The Roland von Glasow Air-Sea-Ice Chamber (RvG-ASIC): an experimental facility for studying ocean/sea-ice/atmosphere interactions" by M. Thomas and co-authors describes the experimental sea ice chamber at the University of East Anglia. The manuscript provides a thorough overview of the design and capabilities of the Chamber and it's attendant infrastructure. A series of experimental test runs have been carried out to benchmark the chamber behavior against mass balance, 1D models and to interrogate the internal consistency of instruments, including the techniques for measuring ice thickness. The manuscript is well-written and clearly

laid out and, in my opinion, does an excellent job of featuring the Chamber and providing future users with valuable metrics they can use to design their experiments and test their results. It is clear that the facility is well-equipped for gas measurements as well as radiation studies – both very exciting and relevant phenomena to polar and sea ice research. The benchmark tests and presentation of data are all clear and easy to understand. My only comments have to do with the content and descriptions in Section 2 – the Facility Description. I suggest publication after some moderate revisions to Section 2, to help the reader to conceptualize the facility as it exists.

General comments:

I suggest the authors consider using the passive voice in the description of Section 2 paragraphs where the active voice has been used. Some sentences begin with phrases such as "We use" or "Our version" or "We set up". In general, I am a fan of using the active voice, but in this case, I think it creates the impression of impermanence or haphazard decisions, when in fact, it is clear that both the design and implementation choices are well-thought out.

For example on Line 141, instead of "We use a weather station", the section could begin with "Weather inside the Chamber is measured with a W600-UMB. . ."

Specific comments:

Suggest combining Figures 1 and 3 to make a single unifying diagram of the Schematic in a 3 x 2 panel configuration. Photos could be paired with the diagram that comes closest to revealing that perspective. Common features in the schematic could be annotated in the photos.

For the schematics, I would encourage more use of shading to distinguish the tank from the cold room (as was done in 'view from above') and different line thicknesses to help reveal tank and chamber outlines. Clearly indicate what is the chamber – this refers to the cold room and all its contents? It might be helpful to include some fan

icons and tighten up the arrows and other graphic elements.

Line 199: Do "cold room" and "chamber" refer to the same physical enclosure?

I had some difficulty understanding what was referred to by "chamber" as opposed to "tank" and "cold room". It might be helpful to explicitly define what is encompassed by the word "chamber" in the text and in the combo of Figures 1 and 3.

---

## Author Response (AR1)

We thank the anonymous reviewer and Brice Loose for taking the time to review our manuscript and for their positive comments. We address each comment below. Throughout, our response is in green, the reviewer comments are in black, deletions from the manuscript are in red, and insertions to the manuscript are in blue.

We also added the data sheet for the photodiodes to the SI, which we mistakenly omitted in the first instance, updated the affiliations, added a link to the final version of the supplementary information, and fixed some typos.

Please also see the track changes document that follows our response.

**Response to reviewer 1**

Review of manuscript entitled "The Roland von Glasow Air-Sea-Ice Chamber (RvGASIC): an experimental facility for studying ocean/sea-ice/atmosphere interactions by M Thomas et al. This manuscript describes a state-of-the-art laboratory facility for preparing laboratory grown sea ice in a setting that can be exploited for process study. The manuscript should be of high interest to the scientific community. The text is well written, concise, accurate, and the figures are appropriate. I have no concerns about this manuscript and recommend it be published almost as is. I have only a few, very minor comments and a few questions for the authors:

105: "carrying out measurements" instead of "measuring"?
We have amended the text

The RvG-ASIC has a suite of instruments for  carrying out measurements of the experimental ocean, sea ice, and atmosphere (Table 1).

Fig.2 impossible to distinguish line shades / colors
We have amended the figure

413, 414: "air temperatures" are "high" or "low", not "warm" or "cold"
We have amended the text

What is the thickest ice that can be grown in this facility? The text says it can be 20cm and still be floating. It's not clear whether the ice can be grown thicker?
We have added some more information to the text

the surface floods (e.g. Rysgaard et al., 2014), resulting in a shiny, liquid surface layer. Sea ice, fast to the tank walls, has been grown up to 25 cm thickness, and could potentially be grown thicker. With insufficient side heating and insulation the ocean

How is ice growth prevented in the side tank? I assume the side tank is insulated on top, but it's not stated explicitly.
We have added some information to the text

joined to the main tank, connected by four 100 mm holes (Figure 1). This side tank – capped with a lid – is never allowed to freeze over entirely and so provides a path for sample lines into the ocean, a path for cables that does not interfere with the sea-ice/atmosphere interface, and a free path for water displaced by volume expansion upon freezing. The main and side

When sea ice grows, brine rejection at the growing interface necessarily increases the salinity of the ocean, or in this case, in the tank water. I wonder if the side tank in this laboratory setup could be used to help ameliorate this shortcoming associated with a finite-depth tank?
This is a good idea and a possible use of the side tank. However, we have not explored this possibility and feel it is too early for us to say something useful about this.

**Response to Brice Loose (reviewer 2)**
Overview: The manuscript "The Roland von Glasow Air-Sea-Ice Chamber (RvG-ASIC): an experimental facility for studying ocean/sea-ice/atmosphere interactions" by M. Thomas and co-authors describes the experimental sea ice chamber at the University of East Anglia. The manuscript provides a thorough overview of the design and capabilities of the Chamber and it's attendant infrastructure. A series of experimental test runs have been carried out to benchmark the chamber behavior against mass balance, 1D models and to interrogate the internal consistency of instruments, including the techniques for measuring ice thickness. The manuscript is well-written and clearly laid out and, in my opinion, does an excellent job of featuring the Chamber and providing future users with valuable metrics they can use to design their experiments and test their results. It is clear that the facility is well-equipped for gas measurements as well as radiation studies – both very exciting and relevant phenomena to polar and sea ice research. The benchmark tests and presentation of data are all clear and easy to understand. My only comments have to do with the content and descriptions in Section 2 – the Facility Description. I suggest publication after some moderate revisions to Section 2, to help the reader to conceptualize the facility as it exists.

I suggest the authors consider using the passive voice in the description of Section 2 paragraphs where the active voice has been used. Some sentences begin with phrases such as "We use" or "Our version" or "We set up". In general, I am a fan of using the active voice, but in this case, I think it creates the impression of impermanence or haphazard decisions, when in fact, it is clear that both the design and implementation choices are well-thought out. For example on Line 141, instead of "We use a weather station", the section could begin with "Weather inside the Chamber is measured with a W600-UMB. . ."

We have amended the text in several places to remove the active voice from section 2.

et al. (2012) for pure NaCl.  A sonar (Aquascat 1000R)  measures the position of the waterline at the start of the experiment and the position of the base of the sea ice throughout the experiment. We use the GSW toolbox to

 Temperature, $\theta$, profiles through the ocean and sea ice are measured using chains of digital thermometers (Table 1). These have a resolution of $\frac{1}{16}$ °C and are calibrated against $\theta$ measured

 The temperature, wind speed, and relative humidity of our atmosphere are measured using a weather station (WS600-UMB). Two Los Gatos Research (LGR) greenhouse gas analysers measure $CO_2$,

Specific comments: Suggest combining Figures 1 and 3 to make a single unifying diagram of the Schematic in a 3 x 2 panel configuration. Photos could be paired with the diagram that comes closest to revealing that perspective. Common features in the schematic could be annotated in the photos. For the schematics, I would encourage more use of shading to distinguish the tank from the cold room (as was done in 'view from above') and different line thicknesses to help reveal tank and chamber outlines. Clearly indicate what is the chamber – this refers to the cold room and all its contents? It might be helpful to include some fan icons and tighten up the arrows and other graphic elements.

We have amended the schematic following these useful comments. See below:

[Figure]

**Figure 3.** To scale schematic diagram of the coldroom. The three panels show orthogonal views from different vantage points. Crosses and dots indicate air flow away from and towards the viewer, respectively. The lights, shown in grey, are made up of eight sets of visible, UV-A, and UV-B triplets. The main and side tanks are pale green.

After some thought and a few attempts, we chose not to merge figures 1 and 3. Our main reason was that the photos in figure 1 do not correspond to the panels in figure 3, and we felt combining the figures might therefore cause some confusion. A secondary consideration was that, in a combined figure, it is difficult to show the three photos and the schematic as large as we would like. We have added labels to figure 1 (see below) that correspond to figure 3.

[Figure]

[Figure]

[Figure]

**Figure 1.** The tank just after installation (top), with all the main features in place (middle), and set up for experiments with visible lighting (bottom). The labels are consistent with Figure 3, indicating: a) the main tank, b) the side tank, c) the lights, e) atmosphere sample lines and cables, f) ocean sample lines and cables, and i) the (0, 0) position of our sampling coordinate system.

Please see our response to the next comment for our clarification on what the 'chamber' actually is.

Line 199: Do "cold room" and "chamber" refer to the same physical enclosure? I had some difficulty understanding what was referred to by "chamber" as opposed to "tank" and "cold room". It might be helpful to explicitly define what is encompassed by the word "chamber" in the text and in the combo of Figures 1 and 3.

This is a good point. We have added the following text, in a prominent position at the end of Section 2, to clarify what we mean by the 'chamber'.

[revised manuscript text omitted]

$$F_{\text{C}} = K(T_{\text{s}} - T_{\text{f}})/h_{\text{si}}. \tag{17}$$

400    The vertically-averaged thermal conductivity is $K = K_{\text{i}}(1 - \phi) + K_{\text{br}}\phi$, where the subscripts i and br indicate pure ice and brine, respectively. $T_{\text{f}}$ is given by the  salinity-dependent freezing point of the ocean, and $T_{\text{s}}$ is found by iteratively solving the surface energy balance of the sea ice

$$F_{\text{L}\downarrow} - F_{\text{L}\uparrow} + F_{\text{S}} + F_{\text{C}} = 0 \tag{18}$$

[Figure]

**Figure 11.** Vertical temperature (a and c) and bulk salinity profiles (b and d) for two experimental runs. In a and c the model temperature shows the model forcing, which was produced using measured temperature profiles; b and d show salinity estimated from cores, slabs (Cottier et al., 1999), wireharps (Notz et al., 2005), and a gravity drainage model. The horizontal box length shows the median $1\sigma$ standard deviation from repeat measurements at a given depth (cores and slabs) and the vertical box length shows the depth covered by the sample layer. Wireharp errors were calculated using methodology presented in Zeigermann (2018). The model used the Griewank and Notz (2013) gravity drainage parameterisation with tuning parameters taken from Griewank and Notz (2015). The solid line represents their best estimate tuning parameter set. The dotted bounds on the model show output using two other plausible sets of tuning parameters presented in Griewank and Notz (2015).

[Figure]

**Figure 12.** Thickness measured from temperature profiles during the experiments presented in Section 3.1.3 and modelled using Equation 16.

where sensible and latent heat fluxes have been neglected because the wind speed in our experiments was less than 0.1 m s$^{-1}$.

405 We parameterised the downwelling longwave, $F_{\mathrm{L\downarrow}}$, following Efimova (1961), as

$$F_{\mathrm{L\downarrow}} = \sigma(0.746 + 0.0066(\psi/\mathrm{mbar}))T_{\mathrm{
[revised manuscript text omitted]